# TreePiece: Faster Semantic Parsing via Tree Tokenization

**Sid Wang**
Meta Inc. USA
yuwang2020@meta.com

**Akshat Shrivastava**
Meta Inc. USA
akshats@meta.com

**Aleksandr Livshits**
Meta Inc. USA
alll@meta.com

## Abstract

*Autoregressive* (AR) encoder-decoder neural networks have proved successful in many NLP problems, including *Semantic Parsing* – a task that translates natural language to machine-readable *parse trees*. However, the sequential prediction process of AR models can be slow. To accelerate AR for semantic parsing, we introduce a new technique called *TreePiece* that tokenizes a parse tree into subtrees and generates one subtree per decoding step. On TOPv2 benchmark, TreePiece shows 6.1 times faster decoding speed than standard AR, and comparable speed but significantly higher accuracy compared to *Non-Autoregressive* (NAR).

## 1 Introduction

*Autoregressive* (AR) modeling (Sutskever et al., 2014) is a commonly adopted framework in NLP where the next prediction is conditioned on the previously generated tokens. This paper focuses on AR approach for *Semantic Parsing* (Wong, 2005), an NLP task that converts a natural language utterance to a machine-interpretable symbolic representation called *logical form*. The sequence of actions to derive a logical form is isomorphic to a directed tree and often referred to as a *parse tree* (Zettlemoyer and Collins, 2005).

The runtime latency of AR linearly correlates to the output length and could result in low inference speed (Gu et al., 2017; Wang et al., 2018). *Non-Autoregressive* (NAR) modeling (Gu et al., 2017; Wei et al., 2019; Ma et al., 2019), on the other hand, is able to produce outputs in parallel and reduce latency by an order of magnitude (Ghazvininejad et al., 2019). However, NAR performs considerably worse than its AR counterparts without extra training recipes (Wang et al., 2019; Zhou and Keung, 2020; Su et al., 2021). The quality benefits of AR models therefore motivates us to improve their speed, rather than exploring NAR.

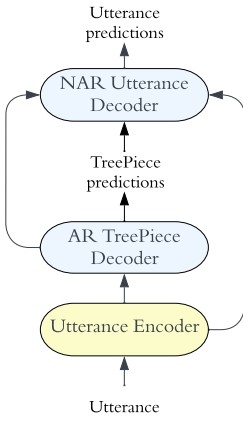

Figure 1: TreePiece-based parse tree modeling design.

**Our contributions**

• We propose a novel approach of tokenizing parse trees into large units called *TreePiece units*, and then building an AR model that predicts one *TreePiece unit* at a time, thus reducing the number of steps needed to generate a full parse tree. To the best of our knowledge, we are the first to extend subword-tokenizer algorithm to semantic trees such that each token is a subtree.

• We validate our approach on TOPv2 benchmark and show that *TreePiece* decoding is 6.1 times faster than standard AR with less than 0.15% accuracy degradation, and nearly as fast as NAR with up to 0.7% accuracy gains.

• We provide theoretical proofs to support our main algorithms and their variants.

## 2 Methods

### 2.1 Parse tree

In this paper, we utilize the *hierarchical semantic representations* based on *intent* and *slot* (Gupta et al., 2018), allowing for modeling complex compositional queries in task-oriented dialog systems. See Figure 2 (LHS) for an example. Now let us

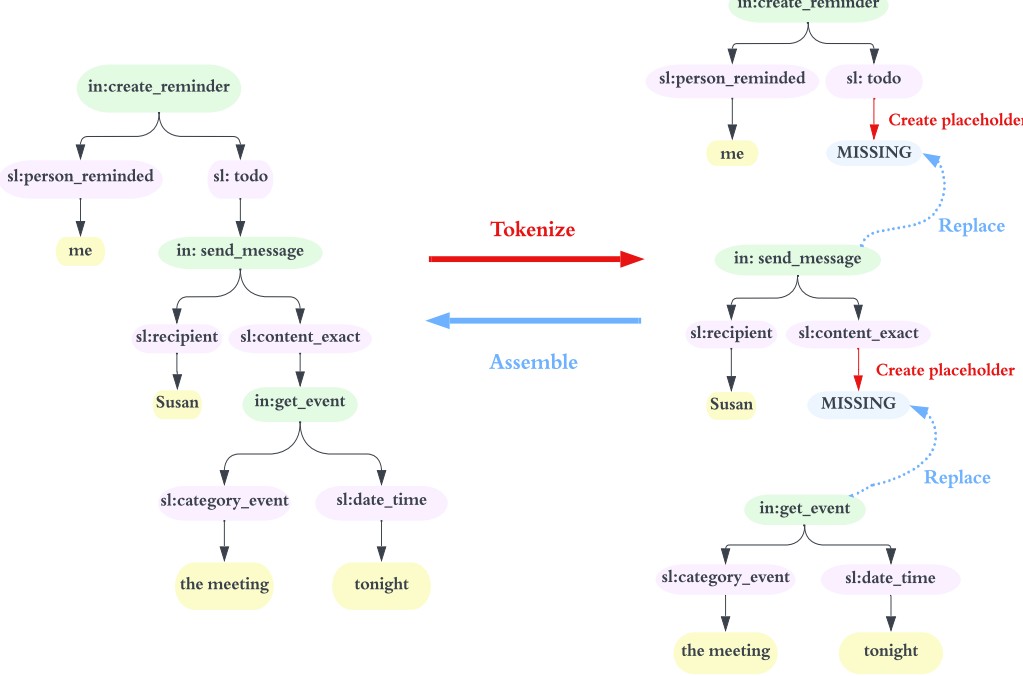

Figure 2: Illustration of tokenizing parse tree/assembling TreePiece units with the placeholder design for given utterance "*Remind me to send Susan an email about the meeting tonight*".

define a few recurring notions in the paper:

**Definition 2.1** (*Ontology*)**.** A *parse tree* node is called an *ontology* iff it represents an *intent/slot*, prefixed by in: and sl: respectively.

**Definition 2.2** (*Skeleton*)**.** The *skeleton* of a *parse tree* is the subtree that consists of all *ontologies*.

**Definition 2.3** (*Utterance leaf*)**.** A *text-span* node is called an *utterance leaf* iff its parent is a *slot*[1].

### 2.2 TreePiece tokenizer

#### 2.2.1 Tokenizer algorithm

**Definition 2.4.** *TreePiece tokenizer* is an algorithm that segments any *skeleton* into subtrees of an open vocabulary. The minimal open vocabulary of a *TreePiece tokenizer* is called a *TreePiece vocabulary*, where an element is called a *TreePiece unit*.

**Definition 2.5** (*TreePiece simplex*)**.** Let $\mathcal{V}$ be a *TreePiece vocabulary* and $t$ be any *TreePiece unit*. A *TreePiece simplex* $\boldsymbol{p}$ is a mapping from $\mathcal{V}$ to the unit interval $[0, 1]$ such that $\sum_{t \in \mathcal{V}} \boldsymbol{p}(t) = 1$.

We propose Algorithm 1 as our *TreePiece tokenizer*, a Viterbi-type (Viterbi, 1967) forward-backward (Nagata, 1994a) algorithm which

computes the optimal tokenization and probability for given skeleton $S$.

**Notations in Algorithm 1**:

(1) $\mathcal{T}$ is the set of all subtrees of $S$ that share the same root as $S$ denoted by $\mathcal{T}$; $\mathcal{T}$ admits a natural filtration $\mathcal{T}_0 \subseteq \cdots \subseteq \mathcal{T}_{d-1} \subseteq \mathcal{T}_d \subseteq \cdots \subseteq \mathcal{T}$, where $\mathcal{T}_d$ is the set of all depth-$d$-subtrees.

(2) $\mathscr{L}$ is the log probability on $\mathcal{T}$ as follows:

$$\mathscr{L}(t) = \begin{cases} \log \boldsymbol{p}(t) & \text{if } t \in \mathcal{V}, \\ -\infty & \text{otherwise} \end{cases}$$

where $\mathcal{V}$ is the *TreePiece vocabulary* and $\boldsymbol{p}$ the *TreePiece simplex*.

(3) for efficiency we apply $\mathsf{Filter}(\cdot, t)$ to restrict to subtrees $t'$ such that (a) $t'$ is a subtree of $t$, (b) the set difference $t'\Delta t$ has exactly one connected component and it is a *TreePiece unit*.

In summary, the *forward* step uses dynamic programming inductive on tree-depth to update all subtrees' log-probabilities and eventually obtain $\mathbb{P}(S; \boldsymbol{p})$ – the probability of the skeleton $S$. The *forward* step also returns a map $\mathscr{P}$ that stores for each $t \in \mathcal{T}$ the optimal position of its previous partition. Then in the *backward* step we can backtrack along the path $S, \mathscr{P}(S), \mathscr{P}(\mathscr{P}(S)), \cdots$ to recover the optimal partition $\pi_S(\boldsymbol{p})$.

---

[1]We adopt the *decoupled form* proposed in (Aghajanyan et al., 2020), which simplifies compositional representations by ignoring text spans that are not *utterance leaves*.

**Algorithm 1** Forward-backward algorithm

**Input:** *TreePiece vocabulary* $\mathcal{V}$, *TreePiece simplex* $\boldsymbol{p}$, and *skeleton* $S$.
**Output:** Partition $\pi_S(\boldsymbol{p})$ and probability $\mathbb{P}(S; \boldsymbol{p})$.
  $\mathcal{T} \leftarrow$ All subtrees of $S$ with the same root.
  $\mathscr{L} \leftarrow \mathsf{Log}(\boldsymbol{p})$
  $\mathscr{P} \leftarrow$ Constant map from $\mathcal{T}$ to BOS token
  $d_{\max} \leftarrow$ Depth of S
  **for** $d = 1, 2, \cdots, d_{\max}$ **do** // Forward begins
    **for** $\mathsf{t} \in \mathcal{T}_d$ **do**
      **for** $d' = 1, 2, \cdots, d$ **do**
        **for** $\mathsf{t}' \in \mathsf{Filter}(\mathcal{T}_{d'}, \mathsf{t})$ **do**
          $\Delta^* \leftarrow \mathsf{t}' \Delta \mathsf{t}$
          $L^* \leftarrow \mathscr{L}(\mathsf{t}') + \sum_{\tau \in \Delta^*} \log \boldsymbol{p}(\tau)$
          **if** $L^* > \mathscr{L}(\mathsf{t})$ **then**
            $\mathscr{L}(\mathsf{t}) \leftarrow L^*, \mathscr{P}(\mathsf{t}) \leftarrow \mathsf{t}'$
  $\mathbb{P}(S; \boldsymbol{p}) \leftarrow \exp(\mathscr{L}(S))$ // Forward ends
  $\mathsf{t}_{\mathrm{curr}} \leftarrow S, \pi_S(\boldsymbol{p}) \leftarrow \emptyset$ // Backward begins
  **while** $\mathsf{t}_{\mathrm{curr}} \neq$ BOS token, **do**
    $\mathsf{t}_{\mathrm{prev}} \leftarrow \mathscr{P}(\mathsf{t}_{\mathrm{curr}}), \Delta^* \leftarrow \mathsf{t}_{\mathrm{prev}} \Delta \mathsf{t}_{\mathrm{curr}}$
    $\pi_S(\boldsymbol{p}) \leftarrow \pi_S(\boldsymbol{p}) \bigcup \Delta^*, \mathsf{t}_{\mathrm{curr}} \leftarrow \mathsf{t}_{\mathrm{prev}}$
  $\pi_S(\boldsymbol{p}) \leftarrow \pi_S(\boldsymbol{p}) \bigcup \{\mathsf{t}_{\mathrm{curr}}\}$ // Backward ends
  **return** $\pi_S(\boldsymbol{p}), \mathbb{P}(S; \boldsymbol{p})$

---

**Algorithm 2** EM algorithm

  Choose $N_0 \in \mathbb{N}^+, \epsilon_0 > 0$; initialize $i \leftarrow 0$, $\Delta \leftarrow +\infty, \mathcal{L}_{\mathrm{prev}} \leftarrow -\infty$.
  **while** $i < N_0$ and $\Delta > \epsilon_0$ **do**
    $\mathcal{L}_{\mathrm{curr}} \leftarrow 0, \mathcal{F}^* \leftarrow$ Zero function on $\mathcal{V}$
    **for** $S \in \mathscr{S}$ **do**
      Compute $\pi_S(\boldsymbol{p}_i)$ and $\mathbb{P}(S; \boldsymbol{p}_i)$   ▷ E-step
      $\mathcal{L}_{\mathrm{curr}} \leftarrow \mathcal{L}_{\mathrm{curr}} + \log \mathbb{P}(S; \boldsymbol{p}_i)$
      **for** $t \in \pi_S(\boldsymbol{p}_i)$ **do**
        $\mathcal{F}^*(t) \leftarrow \mathcal{F}^*(t) + 1$
    **for** $t \in \mathcal{V}$ **do**
      $\boldsymbol{p}_{i+1}(t) \leftarrow \frac{\mathcal{F}^*(t)}{\sum_{\tau \in \mathcal{V}} \mathcal{F}^*(\tau)}$   ▷ M-step
    $i \leftarrow i + 1, \Delta \leftarrow \mathcal{L}_{\mathrm{curr}} - \mathcal{L}_{\mathrm{prev}}$
    $\mathcal{L}_{\mathrm{prev}} \leftarrow \mathcal{L}_{\mathrm{curr}}$

---

### 2.2.2 Tokenizer training

The performance of Algorithm 1 relies on the quality of *TreePiece vocabulary* $\mathcal{V}$ and *TreePiece simplex* $\boldsymbol{p}$. To improve the quality of $\mathcal{V}$ and $\boldsymbol{p}$, we propose a two-stage training procedure:

**Stage 1, Generate $\mathcal{V}$:** Let $\mathscr{S}$ represent all *skeletons* of a given training corpus. Similar to *Byte Pair Encoding* (BPE) (Gage, 1994; Sennrich et al., 2015), we obtain the *TreePiece vocabulary* $\mathcal{V}$ and map $\mathcal{F}_0$ between *TreePiece units* and their frequencies in $\mathscr{S}$. For details see Appendix A.

**Stage 2, Update $\boldsymbol{p}$:** initialize the *TreePiece simplex* $\boldsymbol{p}_0$ as the normalized frequency $\boldsymbol{p}_0(t) =:$ $\mathcal{F}_0(t) / \sum_{\tau \in \mathcal{V}} \mathcal{F}_0(\tau)$ for all $t \in \mathcal{V}$ and then solve for $\boldsymbol{p}_{i+1}$ iteratively as follows:

$$\boldsymbol{p}_{i+1} = \mathrm{argmax}_{\boldsymbol{p}} \sum_{S \in \mathscr{S}} \mathbb{E}_{\Pi_S} \left[ \log \mathbb{P}(S, \pi; \boldsymbol{p}) \big| S; \boldsymbol{p}_i \right]. \tag{1}$$

In general, problem (1) is NP-hard as it involves summing over $\Pi_S$, the set of all possible partitions $\pi$ of a skeleton $S$:

$$\mathbb{E}_{\Pi_S} \left[ \log \mathbb{P}(S, \pi; \boldsymbol{p}) \big| S; \boldsymbol{p}_i \right]$$
$$= \sum_{\pi \in \Pi_S} \log \mathbb{P}(S, \pi; \boldsymbol{p}) \cdot \frac{\mathbb{P}(S, \pi; \boldsymbol{p}_i)}{\mathbb{P}(S; \boldsymbol{p}_i)}. \tag{2}$$

To solve (1) in polynomial time, we propose Algorithm 2 (whose *E-step* uses Algorithm 1) and impose the following assumption on the joint distribution of $S$ and $\pi$:

$$\mathbb{P}(S, \pi; \boldsymbol{p}) \propto \begin{cases} \prod_{\tau \in \pi} \boldsymbol{p}(\tau) & \text{if } \pi = \pi_S(\boldsymbol{p}) \\ 0 & \text{otherwise,} \end{cases} \tag{3}$$

where $\pi_S(\boldsymbol{p}) = \mathrm{argmax}_{\pi \in \Pi_S} \prod_{\tau \in \pi} \boldsymbol{p}(\tau)$. Applying (3), we see that all but one summand in (2) vanish. The following Theorem claims that Algorithm 2 solves for (1) under assumption (3):

**Theorem 2.6.** *Let* $\boldsymbol{p}_0, \boldsymbol{p}_1, \cdots$ *be TreePiece simplices obtained from Algorithm 2. If (3) is true, then (1) holds. Moreover,* $\sum_{S \in \mathscr{S}} \log \mathbb{P}(S; \boldsymbol{p}_i)$ *is monotonically non-decreasing in* $i$.

For proof of Theorem 2.6, see Appendix B.

### 2.3 Modeling

We describe the model that generates *parse tree* components and the method to piece them together.

### 2.3.1 Modeling mechanism

As illustrated in Figure 1, an encoder computes the hidden states of a given utterance, then an AR decoder consumes the encoder hidden states and generate *TreePiece units* autoregressively. The technique in Subsection 2.3.2 will allow us to put these units together and obtain a full *skeleton*. The *skeleton* then uniquely determines the number (denoted by $N$) and positions of all utterance leaves (see Figure 2), which offers us the convenience to use an NAR decoder to generate all utterance leaves within one step. For NAR utterance decoder, we closely follow (Ghazvininejad et al., 2019): first

|  | EM (%) | EM-S (%) | CPU Decoding time (ms) | CPU Inference time (ms) | GPU Decoding time (ms) | GPU Inference time (ms) |
|---|---|---|---|---|---|---|
| *Baselines* | | | | | | |
| AR | **86.99** | 89.13 | 45.53 | 63.81 | 44.87 | 55.77 |
| NAR | 86.29 | 88.56 | **7.00** | **25.21** | **6.42** | **17.56** |
| *Our method* | | | | | | |
| TreePiece-1200 | 86.51 | 89.05 | 7.61 | 25.89 | 6.66 | 17.75 |
| TreePiece-800 | 86.73 | 89.13 | 7.79 | 26.04 | 7.09 | 18.12 |
| **TreePiece-600** | 86.86 | **89.26** | 7.85 | 26.14 | 7.34 | 18.45 |
| TreePiece-500 | 86.65 | 89.10 | 7.94 | 26.53 | 7.81 | 19.84 |
| TreePiece-400 | 86.56 | 89.02 | 7.86 | 26.21 | 7.91 | 19.97 |
| TreePiece-300 | 86.47 | 89.04 | 8.09 | 26.79 | 7.94 | 19.83 |
| TreePiece-200 | 86.51 | 89.02 | 8.09 | 26.57 | 8.19 | 20.18 |
| TreePiece-100 | 86.57 | 89.05 | 8.51 | 27.19 | 8.65 | 20.60 |
| TreePiece-0 | 86.31 | 88.56 | 11.57 | 29.73 | 12.01 | 24.02 |

Table 1: Quality and latency of all models on TOPv2. We train each model with 3 random seeds, and report the averaged EM/EM-S scores and latency on test split of TOPv2 dataset. We measure the decoding and overall inference latency of all models on both CPU and NVIDIA V100 32 GB GPU, and report the averaged milliseconds over all test samples. The number suffix for a TreePiece model represents the expansion size when creating TreePiece vocabulary. The best entry for each metric is bolded.

prepare the NAR decoder's input by concatenating the embeddings of the predicted TreePieces and $N$ mask tokens. Then each decoder layer performs self-attention as well as cross attention with encoder hidden states. Lastly the decoder generates utterance predictions at these $N$ masked positions.

### 2.3.2 Assemble TreePiece units

Unlike subword-tokenization, where original sentence can be trivially recovered from subword units via string concatenation, there is no canonical way to reassemble *TreePiece units*. To overcome this issue, we allow *TreePiece units* to have placeholders[2], and require that two units can only be joined at a placeholder node. This design provides a unique way to glue a sequence of ordered (e.g. pre/level-ordered) *TreePiece units*, as shown in Figure 2.

## 3 Experiments

### 3.1 Datasets

We train, validate, and test our approach on the publicly available benchmark TOPv2 (Chen et al., 2020), a multi-domain task-oriented semantic parsing dataset. The dataset provides a training/validation/test split. Throughout our experi-

---

[2]Finding all possible placeholder patterns is NP-hard and unnecessary. In Appendix D we provide a practical solution.

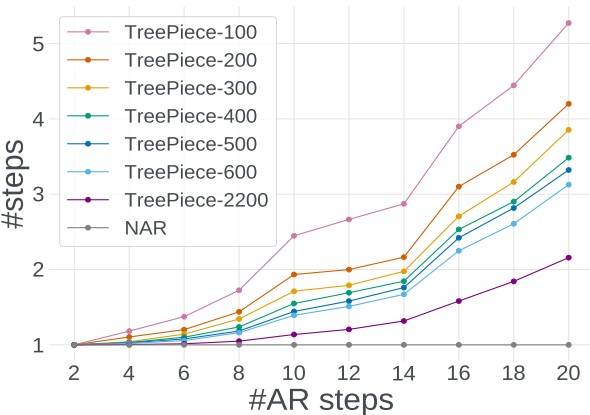

Figure 3: Plot of averaged TreePiece decoding steps against AR decoding steps for skeleton generations.

ments, we use the training split to train the models, the validation split for earlystopping, model checkpointing, and hyperparameter tuning, and the test split to report the best model's performance.

### 3.2 Metrics

We evaluate the model performance on two metrics: *Exact Match* (EM) respectively *Exact Match of Skeleton* (EM-S), defined to be the percentage of utterances whose *logical forms* respectively *skeletons* are correctly predicted (Shrivastava et al., 2022).

## 3.3 Baselines

We compare our approach against 2 baselines: AR and NAR. Both baselines are sequence-to-sequence (seq2seq) that produces subword units of serialized logical forms. Their output space consists of *ontologies* (prefixed by left bracket "["), *utterance leaves*[3], and right bracket "]".

**AR baseline** admits a standard AR structure. It has an *autoregressive* decoder that generates serialized logical forms by producing one token at a time.

**NAR baseline** adopts *mask-predict* (Ghazvininejad et al., 2019) with beam size 1, which predicts the output length first and then generates all tokens in one step using a *non-autoregressive* decoder.

## 3.4 Experiment setup

For TreePiece models, we experiment with 9 different expansion sizes (used in **Stage 1**, Subsection 2.2.1) varying from 0 to 1200. We optimize the hyperparameters for both baselines and TreePiece-600, and apply the same hyperparameters from TreePiece-600 to all other TreePiece models. We defer the model configurations, training details, hyperparameter choices to Appendix E.

## 4 Results

### 4.1 Quality

As shown in Table 1, TreePiece model sees up to 0.7% relative improvements over NAR and less than 0.15% degradation from AR in terms of EM, while achieving the best EM-S score among all approaches, especially showing 0.8% relative improvement over NAR. We attribute TreePiece's high quality on skeleton predictions to its ability to respect the tree structure of logical forms and generating 100% valid outputs by design so that the model can better focus on utterance-understanding without being distracted by any structure issue.

Table 2 further shows TreePiece's privilege over NAR in handling tasks of higher complexity, achieving $> 2\%$ improvements for frames with more than 1 intents.

### 4.2 Latency

Table 1 indicates that TreePiece makes decoding 6.1\5.8 times faster and overall inference 3.0\2.5 times faster than AR on GPU\CPU, with only 5% inference latency growth compared to NAR. In

---

[3]We represent *utterance leaves* in *span-pointers* (Shrivastava et al., 2021) form to simplify the parsing task.

| #Intents | | 1 | 2 | 3 | 4 |
|---|---|---|---|---|---|
| TreePiece | EM | 88.10 | 83.21 | 69.46 | 46.51 |
| -600 | EM-S | 90.01 | 87.23 | 72.73 | 50.00 |
| NAR | EM | 87.72 | 80.95 | 67.60 | 23.26 |
| | EM-S | 89.65 | 84.81 | 69.93 | 24.42 |

Table 2: Comparison on TOPv2 tasks with different level of complexity in terms of number of intents.

Figure 3, we compare the decoding steps needed to generate full skeletons between TreePiece decoder (of 7 different expansion sizes) and AR decoder. The plot illustrates the acceleration effects of our approach, showing that TreePiece with just 200 expansion-size can already reduce the averaged decoding steps by 83.3% compared to AR.

**Related work** *Autoregressive* modeling have been used in a range of *Semantic Parsing* works (Tai et al., 2015; Cheng et al., 2017b; Dong and Lapata, 2018). Especially, the Sequence-to-Tree scheme was adopted by (Dong and Lapata, 2016). To speed up the inference time, *Non-autoregressive* modeling were introduced to the field of *Machine Translation* (Gu et al., 2017; Lee et al., 2018; Libovický and Helcl, 2018), and later become popular in *Semantic Parsing* as well (Ghazvininejad et al., 2019; Babu et al., 2021; Shrivastava et al., 2021). However, to match the quality of AR, extra training stages are necessary such as *Knowledge Distillation* from AR models (Gu et al., 2017; Lee et al., 2018; Wei et al., 2019; Stern et al., 2019). On the other hand, (Rubin and Berant, 2020) improves AR decoding's efficiency via *Bottom-Up Parsing* (Cheng et al., 2017a). Our paper takes a completely different path from all previous work by extending the subword tokenization algorithms (Nagata, 1994b; Scott, 2002; Sennrich et al., 2015; Kudo, 2018; Kudo and Richardson, 2018) to trees.

## Conclusion

This paper proposes a novel way to model and speed up *Semantic Parsing* via tokenizing parse trees into subtrees. We provide thorough elucidations and theoretical supports for our technique, and demonstrate significant improvements in terms of speed and quality over common AR and NAR baselines on the TOPv2 benchmark.

## Limitations

The proposed TreePiece technique, while evaluated on TOPv2 dataset, is not intrinsically bound to it. Indeed, our approach requires only two conditions on a dataset for applicability:

- offers a closed vocabulary of ontologies;

- logical forms inherently carry tree structures.

As a matter of fact, TreePiece can seamlessly adapt to a broad range of datasets, including Wik-iSQL (Zhong et al., 2017), WEBQUESTIONS (Berant et al., 2013), SequentialQA (Iyyer et al., 2017), GEOquery (Davis and Meltzer, 2007), Spider (Yu et al., 2018), ATIS (Hemphill et al., 1990), etc. Despite this, we solely focused on showcasing its effectiveness in the specific case of "task-oriented natural language understanding based on intent and slots". Additionally, our approach employs standard autoregressive decoding for sub-tree generation, neglecting the exploration of even more efficient decoding techniques. Lastly, our current tokenization algorithm may introduce out-of-vocabulary (OOV) tokens; while we proposed effective ways to reduce OOV rates, it however cannot fully eliminate the OOV phenomena.

## Ethics Statement

Our proposed method presents a novel tokenization operation for tree-like data, yielding substantial practical implications in semantic parsing domains such as natural language understanding, SQL generation, code generation, etc. However, it is crucial to acknowledge that, similar to many other tokenization algorithms, our approach may introduce biases from the training data into vocabulary and tokenization patterns. Consequently, practitioners needs to be mindful of this when curating the training corpus before utilizing our method.

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

# A   Appendix: Vocabulary generation

This stage resembles the merging operation in *Byte Pair Encoding* (BPE) (Gage, 1994; Sennrich et al., 2015). Given a training corpus, denote its skeletons by $\mathscr{S}$. We initialize the *TreePiece vocabulary* $\mathcal{V}$ as the set of *ontologies* extracted from $\mathscr{S}$ and $\mathcal{F}_0$ as the map between *ontologies* and their frequencies in $\mathscr{S}$. Now repeat the steps below until $\mathcal{V}$ reaches a pre-determined size:

- Count the frequencies of all adjacent but unmerged *TreePiece unit* pairs in $\mathscr{S}$. Find the most frequent pair $p^*$ and its frequency $n^*$.

- Merge $p^*$ in every $S \in \mathscr{S}$ that contains $p^*$, add $p^*$ to $\mathcal{V}$, and update $\mathcal{F}_0$ with $\mathcal{F}_0(p^*) = n^*$.

# B   Appendix: Proof of Theorem 2.6

For convenience we adopt the following notations.

**Notation B.1.** Let $\boldsymbol{p}$ be a *TreePiece simplex* and $\pi =: [\tau_1, \cdots, \tau_k]$ be a partition where each $\tau_i$ is a *TreePiece unit*. Define $\boldsymbol{p}(\pi) =: \prod_{\tau \in \pi} \boldsymbol{p}(\tau)$.

**Notation B.2.** Let $\pi =: [\tau_1, \cdots, \tau_k]$ be a partition and $\tau$ be any *TreePiece unit*. Define $n(\pi, \tau) =: \sum_{\tau_i \in \pi} \mathbb{1}_{\tau = \tau_i}$. In other words, $n(\pi, \tau)$ is the number of appearances of $\tau$ in $\pi$.

Now we introduce a general hypothesis and will prove a key lemma under this hypothesis.

**Hypothesis B.1.** *The joint distribution of skeleton $S$ and partition $\pi$ satisfies the following rule,*

$$\mathbb{P}(S, \pi; \boldsymbol{p}) \propto \begin{cases} \prod_{\tau \in \pi} \boldsymbol{p}(\tau) \cdot \chi(\pi, \boldsymbol{p}) \text{ if } \pi \in \Pi_S \\ 0 \text{ otherwise,} \end{cases}$$
(4)

*where $\chi : \Pi_S \times [0,1]^{|\mathcal{V}|} \to \{0, 1\}$ is locally smooth almost everywhere (under Lebesgue measure on $[0,1]$). In other words, for a.e. $\boldsymbol{p} \in [0,1]^{|\mathcal{V}|}$ and every $\pi \in \Pi_S$ there exists a neighborhood $B_\epsilon(\boldsymbol{p})$ where $\chi(\pi, \cdot)$ is constant.*

**Remark B.1.** *Assumption* (3) *is a special case of hypothesis B.1, where $\chi(\pi, \boldsymbol{p}) = 1$ if $\pi = \pi_S(\boldsymbol{p})$ and $0$ otherwise.*

**Remark B.2.** *Without loss of generality, in equations* (3) *and* (4) *we replace the symbol "$\propto$" with "$=$", which otherwise complicates all formulae expressions with a non-essential scalar constant.*

**Lemma B.1.** *Under Hypothesis B.1, $\boldsymbol{p}_{k+1}$ is a solution to* (1) *iff the following holds $\forall \tau^* \in \mathcal{V}$ :*

$$\boldsymbol{p}_{k+1}(\tau^*) = \frac{\sum_{S \in \mathscr{S}} \mathbb{E}_{\Pi_S}[n(\pi, \tau^*)|S; \boldsymbol{p}_k]}{\sum_{\tau \in \mathcal{V}} \sum_{S \in \mathscr{S}} \mathbb{E}_{\Pi_S}[n(\pi, \tau)|S; \boldsymbol{p}_k]}.$$
(5)

*proof of Theorem 2.6 assuming Lemma B.1 holds.* By following the *E-step* of Algorithm 2, we can express the frequency $\mathcal{F}^*(t)$ as

$$\sum_{S \in \mathscr{S}} \sum_{\tau \in \pi_S(\boldsymbol{p}_i)} \mathbb{1}_{\tau = t} = \sum_{S \in \mathscr{S}} n(\pi_S(\boldsymbol{p}_i), t).$$
(6)

Assumption 3 says that the probability measure $\mathbb{P}(\pi|S; \boldsymbol{p}_k)$ is supported on the singleton $\pi_S(\boldsymbol{p}_i)$, therefore the following holds for all $\tau \in \mathcal{V}$:

$$n(\pi_S(\boldsymbol{p}_i), \tau) = \mathbb{E}_{\Pi_S}[n(\pi, \tau)|S; \boldsymbol{p}_k].$$
(7)

Now inserting the identity (7) to the right hand side of equation (6) we obtain

$$\mathcal{F}^*(t) = \sum_{S \in \mathscr{S}} \mathbb{E}_{\Pi_S}[n(\pi, t)|S; \boldsymbol{p}_k].$$
(8)

Next, Inserting (8) to the *M-step* in Algorithm 2, we have

$$\boldsymbol{p}_{k+1}(t) = \frac{\sum_{S \in \mathscr{S}} \mathbb{E}_{\Pi_S}[n(\pi, t)|S; \boldsymbol{p}_k]}{\sum_{\tau \in \mathcal{V}} \sum_{S \in \mathscr{S}} \mathbb{E}_{\Pi_S}[n(\pi, \tau)|S; \boldsymbol{p}_k]}.$$
(9)

Invoking Lemma B.1, we see that $\boldsymbol{p}_{k+1}$ is the solution to problem (1), which proves the first conclusion in Theorem 2.6.

Secondly, the monotonicity of $\log \mathbb{P}(S; \boldsymbol{p}_i)$ can

be achieved as follows:

$$\sum_{S\in\mathscr{S}} \log \mathbb{P}(S; \boldsymbol{p}_{k+1})$$

$$= \sum_{S\in\mathscr{S}} \log \mathbb{P}(S, \pi_S(\boldsymbol{p}_{k+1}); \boldsymbol{p}_{k+1})$$

$$= \sum_{S\in\mathscr{S}} \boldsymbol{p}_{k+1}(\pi_S(\boldsymbol{p}_{k+1}))$$

$$\geq \sum_{S\in\mathscr{S}} \boldsymbol{p}_{k+1}(\pi_S(\boldsymbol{p}_k))$$

$$= \sum_{S\in\mathscr{S}} \log \mathbb{P}(S, \pi_S(\boldsymbol{p}_k); \boldsymbol{p}_{k+1})$$

$$= \sum_{S\in\mathscr{S}} \mathbb{E}[\log \mathbb{P}(S, \pi_S(\boldsymbol{p}_k); \boldsymbol{p}_{k+1})|S; \boldsymbol{p}_k] \quad (10)$$

$$\geq \sum_{S\in\mathscr{S}} \mathbb{E}[\log \mathbb{P}(S, \pi_S(\boldsymbol{p}_k); \boldsymbol{p}_k)|S; \boldsymbol{p}_k]$$

$$= \sum_{S\in\mathscr{S}} \boldsymbol{p}_k(\pi_S(\boldsymbol{p}_k))$$

$$= \sum_{S\in\mathscr{S}} \log \mathbb{P}(S, \pi_S(\boldsymbol{p}_k); \boldsymbol{p}_k)$$

$$= \sum_{S\in\mathscr{S}} \log \mathbb{P}(S; \boldsymbol{p}_k).$$

Here, all equalities are consequences of Assumption 3, the first inequality follows from the definition of $\pi_S(\boldsymbol{p}_{k+1})$, and the second inequality uses the maximization property of $\boldsymbol{p}_{k+1}$:

$$\boldsymbol{p}_{k+1} = \mathrm{argmax}_{\boldsymbol{p}} \sum_{S\in\mathscr{S}} \mathbb{E}[\log \mathbb{P}(S, \pi_S(\boldsymbol{p}_k); \boldsymbol{p})|S; \boldsymbol{p}_k].$$

(11)

Thus concludes Theorem 2.6. □

To complete the proof of Theorem 2.6 it suffices to prove Lemma B.1.

*proof of Lemma B.1.* Consider the following Lagrange multiplier of problem (1):

$$\mathcal{L}(\boldsymbol{p}, \lambda) = \sum_{S\in\mathscr{S}} \mathbb{E}\big[\log \mathbb{P}(S, \pi; \boldsymbol{p})\big|S; \boldsymbol{p_k}\big]$$
$$+ \lambda(\sum_{\tau\in\mathcal{V}} \boldsymbol{p}(\tau) - 1). \quad (12)$$

Plugging (4) into the above equation, we get

$$\sum_{S\in\mathscr{S}} \sum_{\pi\in\Pi_S} \log \boldsymbol{p}(\pi) \cdot \mathbb{P}(\pi|S; \boldsymbol{p}_k)$$

$$+ \sum_{S\in\mathscr{S}} \sum_{\pi\in\Pi_S} \log \chi(\pi, \boldsymbol{p}) \cdot \mathbb{P}(\pi|S; \boldsymbol{p}_k) \quad (13)$$

$$+ \lambda(\sum_{\tau\in\mathcal{V}} \boldsymbol{p}(\tau) - 1)$$

$$:= \mathrm{I} + \mathrm{II} + \mathrm{III}.$$

Inserting equation (13) to the following identity:

$$\nabla_{\boldsymbol{p}, \lambda} \mathcal{L} = \boldsymbol{0}, \quad (14)$$

we obtain for each $\tau^* \in \mathcal{V}$ that

$$\sum_{S\in\mathscr{S}} \sum_{\pi\in\Pi_S} \frac{n(\pi, \tau^*)}{\boldsymbol{p}(\tau^*)} \cdot \mathbb{P}(\pi|S; \boldsymbol{p}_k) + \lambda = 0. \quad (15)$$

Note the locally constant assumption in Hypothesis B.1 makes the derivative of term II vanishes *a.e.*. Identity (15) then allows us to solve for $\boldsymbol{p}(\tau^*)$:

$$\boldsymbol{p}(\tau^*) = -\frac{1}{\lambda} \cdot \sum_{S\in\mathscr{S}} \sum_{\pi\in\Pi_S} n(\pi, \tau^*) \cdot \mathbb{P}(\pi|S; \boldsymbol{p}_k).$$

(16)

Next, using the simplex property $\sum_{\tau\in\mathcal{V}} \boldsymbol{p}(\tau) = 1$ and summing up (16) over $\mathcal{V}$, we find $\lambda$:

$$-\frac{1}{\sum_{\tau\in\mathcal{V}} \sum_{S\in\mathscr{S}} \sum_{\pi\in\Pi_S} n(\pi, \tau) \cdot \mathbb{P}(\pi|S; \boldsymbol{p}_k)}.$$

(17)

Plugging the above value of $\lambda$ back to (16), we obtain the final expression of $\boldsymbol{p}(\tau^*)$:

$$\frac{\sum_{S\in\mathscr{S}} \sum_{\pi\in\Pi_S} n(\pi, \tau^*) \cdot \mathbb{P}(\pi|S; \boldsymbol{p}_k)}{\sum_{\tau\in\mathcal{V}} \sum_{S\in\mathscr{S}} \sum_{\pi\in\Pi_S} n(\pi, \tau) \cdot \mathbb{P}(\pi|S; \boldsymbol{p}_k)}$$
$$= \frac{\sum_{S\in\mathscr{S}} \mathbb{E}\big[n(\pi, \tau^*)\big|S; \boldsymbol{p}_k\big]}{\sum_{\tau\in\mathcal{V}} \sum_{S\in\mathscr{S}} \mathbb{E}\big[n(\pi, \tau)\big|S; \boldsymbol{p}_k\big]}, \quad (18)$$

which is precisely (5). This proves the *if* direction of the Lemma. Indeed, a maximizer $\boldsymbol{p}$ must be a critical point of the Lagrange multiplier and satisfy (14), therefore identity (18) holds. Conversely, identity (18) for arbitrary $\tau^* \in \mathcal{V}$ fully characterize $\boldsymbol{p}$, and by the *if* direction it can only be the unique maximum. This proves the opposite direction, and completes the proof of Lemma B.1. □

## C   Appendix: An FFBS algorithm

We propose Algorithm 3, a Forward-Filtering Backward-Sampling (FFBS) (Scott, 2002; Kudo, 2018; Kudo and Richardson, 2018) algorithm under the setting of TreePiece. We highlight those lines in Algorithm 3 that differ from Algorithm 1. Their main distinctions lie in (1) update formula for probabilities, (2) backward strategy.

Before *forward*, we call GetInitPairProbs to initialize a probability function on the Cartesian product space $\mathcal{T} \times \mathcal{T} \bigcup \{\text{BOS}\}$ as follows:

$$\mathscr{P}(\mathsf{t}, \mathsf{t}') = \begin{cases} \boldsymbol{p}(\mathsf{t}) & \text{if } \mathsf{t} \in \mathcal{V} \text{ and } \mathsf{t}' = \text{BOS token,} \\ 0 & \text{otherwise.} \end{cases}$$

During *backward*, we call `Sampling` to randomly sample a previous subtree of $\mathfrak{t}$ with respect to the following distribution:

$$\left\{ \frac{\exp(\theta \cdot \log \mathscr{P}(\mathfrak{t}', \mathfrak{t}))}{\sum_{\mathfrak{s} \in \mathcal{T}(\mathfrak{t})} \exp(\theta \cdot \log \mathscr{P}(\mathfrak{s}, \mathfrak{t}))} \right\}_{\mathfrak{t}' \in \mathcal{T}(\mathfrak{t})} \quad (19)$$

where $\mathcal{T}(\mathfrak{t}) =: \{t' \in \mathcal{T} : \mathscr{P}(\mathfrak{t}, \mathfrak{t}') > 0\}$. Here a smaller $\theta$ leads to a more uniform sampling distribution among all partitions, while a larger $\theta$ tend to select the Viterbi partition picked by Algorithm 1 (Kudo, 2018).

---

**Algorithm 3** Forward-Filtering Backward Sampling (FFBS) Algorithm

---

**Input:** TreePiece vocabulary $\mathcal{V}$, TreePiece simplex $\boldsymbol{p}$, skeleton $S$, sampling coefficient $\theta$.
**Output:** Partition $\pi_S(\boldsymbol{p})$ and probability $\mathbb{P}(S; \boldsymbol{p})$.
  $\mathcal{T} \leftarrow$ All subtrees of $S$ with the same root.
  $\mathscr{L} \leftarrow \text{Log}(\boldsymbol{p}), \mathcal{Q} \leftarrow \exp \circ \mathscr{L}$
  $\mathscr{P} \leftarrow \text{GetInitPairProbs}(\boldsymbol{p})$
  $d_{\max} \leftarrow$ Depth of S
  **for** $d = 1, 2, \cdots, d_{\max}$ **do** // Forward begins
    **for** $\mathfrak{t} \in \mathcal{T}_d$ **do**
      **for** $d' = 1, 2, \cdots, d$ **do**
        **for** $\mathfrak{t}' \in \text{Filter}(\mathcal{T}_{d'}, \mathfrak{t})$ **do**
          $\Delta^* \leftarrow \mathfrak{t}' \Delta \mathfrak{t}$
          $Q^* \leftarrow \mathcal{Q}(\mathfrak{t}') \cdot \prod_{\tau \in \Delta^*} \boldsymbol{p}(\tau)$
          $\mathcal{Q}(\mathfrak{t}) \leftarrow \mathcal{Q}(\mathfrak{t}) + Q^*$
          $\mathscr{P}(\mathfrak{t}, \mathfrak{t}') \leftarrow Q^*$
  $\mathbb{P}(S; \boldsymbol{p}) \leftarrow \mathcal{Q}(S)$ // Forward ends
  $\mathfrak{t}_{\text{curr}} \leftarrow S, \pi_S(\boldsymbol{p}) \leftarrow \emptyset$ // Backward begins
  **while** $\mathfrak{t}_{\text{curr}} \neq$ BOS token, **do**
    $\mathfrak{t}_{\text{prev}} \leftarrow \text{Sampling}(\mathscr{P}, \mathfrak{t}_{\text{curr}}, \theta)$
    $\Delta^* \leftarrow \mathfrak{t}_{\text{prev}} \Delta \mathfrak{t}_{\text{curr}}$
    $\pi_S(\boldsymbol{p}) \leftarrow \pi_S(\boldsymbol{p}) \bigcup \Delta^*, \mathfrak{t}_{\text{curr}} \leftarrow \mathfrak{t}_{\text{prev}}$
  $\pi_S(\boldsymbol{p}) \leftarrow \pi_S(\boldsymbol{p}) \bigcup \{\mathfrak{t}_{\text{curr}}\}$ // Backward ends
  **return** $\pi_S(\boldsymbol{p}), \mathbb{P}(S; \boldsymbol{p})$

---

Algorithm 3 allows us to sample from all possible partitions rather than generating fixed patterns. In practice, this version is used in place of Algorithm 1 to reduce the OOV rates; see Appendix D for further discussions.

**Remark C.1.** Let us assume the following holds in place of Assumption (3):

$$\mathbb{P}(S, \pi; \boldsymbol{p}) \propto \begin{cases} \prod_{\tau \in \pi} \boldsymbol{p}(\tau) & \text{if } \pi \in \Pi_S \\ 0 & \text{otherwise,} \end{cases} \quad (20)$$

another special case of Hypothesis B.1 with $\chi(\pi, \boldsymbol{p}) \equiv 1$. By Lemma B.1, solving problem

(1) requires computing $\mathbb{E}_{\Pi_S}[n(\pi, \tau)|S; \boldsymbol{p}_k]$, which now becomes NP-hard. But we can utilize Algorithm 3 to obtain an approximate solution. Indeed, if we iteratively run Algorithm 3 in place of the *E-step* in Algorithm 1 $K$ times to obtain a partition sequence $\pi_S(\boldsymbol{p})^{(1)}, \pi_S(\boldsymbol{p})^{(2)}, \cdots, \pi_S(\boldsymbol{p})^{(K)}$, and use the averaged partitions to update the frequency $\mathcal{F}^*$, then following similar lines in Appendix B, we can prove an asymptotic version of Theorem 2.6 under Assumption 20, by showing that the averaged frequency over $K$ partitions converges to $\mathbb{E}[n(\pi, \tau)|S; \boldsymbol{p}_k]$ as $K$ tends to infinity, a direct consequence of *Law of Large Numbers*. We omit the details.

# D  Appendix

As discussed in Subsection 2.3.2, a placeholder structure is necessary for well-defined assembly of *TreePiece units*. However, adding all possible placeholder patterns to vocabulary is impractical for both time and memory. Instead, we shall include only those patterns that most likely to occur. To do so, we tokenize every training skeleton and add the results to the *TreePiece vocabulary*. As illustrated by the "Tokenize" direction in Figure 2, when a node loses a child during tokenization, we attach a placeholder to the missing child's position.

**Remark D.1.** There may exist new placeholder patterns that are *Out-Of-Vocabulary* (OOV) at inference time. To mitigate OOV, we apply Algorithm 3 (in place of Algorithm 1) to tokenize each training skeleton $K_0$ times. Both $K_0$ and the sampling coefficient $\theta_0$ will be specified in Appendix E.1.2. Intuitively, with a larger $K_0$ and a smaller $\theta_0$, Algorithm 3 is able to generate more abundant placeholder patterns to cover as many OOV placeholders as possible.

# E  Appendix

## E.1  Model configurations

### E.1.1  Model architectures

Across all of our experiments, we use RoBERTa-base encoder (Liu et al., 2019) and transformer (Vaswani et al., 2017) decoders. For encoder architecture, we refer the readers to (Liu et al., 2019). All models' decoder have the same multi-head-transformer-layer architecture (embedding dimension 768, 12 heads). Both AR and NAR's decoders have 2 layers. For TreePiece archtecture, *TreePiece decoder* and *Utterance decoder* has 1 layer each.

| Modules | TreePiece model | | | NAR baseline | | | AR baseline | |
|---|---|---|---|---|---|---|---|---|
| | Encoder | TreePiece decoder | Utterance decoder | Encoder | Length predictor | Decoder | Encoder | Decoder |
| Learning rates | $4 \times 10^{-6}$ | $6 \times 10^{-5}$ | $4 \times 10^{-5}$ | $2 \times 10^{-5}$ | $1 \times 10^{-4}$ | $6 \times 10^{-5}$ | $4 \times 10^{-6}$ | $2 \times 10^{-5}$ |
| Decay coefficients | 0.999 | | | 0.999 | | | 0.9999 | |

Table 3: Optimization hyperparameter choices for all models

**[ intent | create reminder [ slot | person reminded ] [ slot | todo [ missing | [ intent | send message [ slot | recipient ] [ slot | content exact [ missing | [ intent | get event [ slot | category event ] [ slot | date time ] ] ] ] ] ] ] ]**

Figure 4: Serializing the skeleton in Figure 2 (LHS) using the *"placeholder nest"* design.

Note for fairness of comparisons, we let each model have exactly 2 decoder layers in total.

### E.1.2 TreePiece vocabulary

We extract from the TOPv2 dataset 162 ontologies in total, and use TOPv2 training split as training corpus to iteratively run vocabulary generation (ref. Appendix A) 0, 100, 200, 300, 400, 500, 600 times to obtain vocabularies of size 162, 262, 362, 462, 562, 762 respectively. Since we only optimize for TreePiece-600 let us only focus on its vocabulary (of size 762). We then apply the Algorithm 2 with $N_0 = 30$ and $\epsilon_0 = 0.01$ to train the tokenizer and arrive at a vocabulary of 2153 different TreePiece unit patterns with placeholders. Finally, we follow Appendix D (with $K_0 = 10, \theta_0 = 0.15$) and expand the *TreePiece vocabulary* to size 3817. Note the vocabulary obtained this way has less than 0.1% OOV rate on test dataset, compared to 0.45% were we not using the sampling trick in Remark D.1.

### E.2 Training details

### E.2.1 TreePiece embedding

Within the *TreePiece decoder*, we tie the classifer head's weight to the *TreePiece unit* embedding matrix, and found it beneficial to pretrain this weight rather than randomly initializing it. We take inspirations from (Shrivastava et al., 2022) and create the pretraining corpus by serializing all *skeletons* in the training dataset. To let the placeholder information blend into the corpus, we introduce a *placeholder nest* structure and add it to the serialized logical forms, as illustrated by Figure 4. Finally, we use the masked language model (MLM) (Devlin et al., 2019) pre-training objective with mask-rate 0.15

and train for up to 20 epochs until convergence.

### E.2.2 Hyperparameter choices

Across all experiments, we set the batch size to be 256, and total number of epochs to be 100 with early stopping when validation EM (ref. Subsection 3.2) stops improving. For optimization, we use Adam optimizer (Kingma and Ba, 2014) with weight decay 0.01 and $\epsilon = 10^{-8}$. In addition, we warm up the learning rate in 5 epochs and then exponentially decay (Senior et al., 2013) at the end of every epoch.

We also observe that each module favors learning rates with different magnitude, so we do learning rate search separately for each module among the interval $[10^{-6}, 10^{-4}]$. For exponential decay coefficients we optimize them among $\{0.9, 0.99, 0.999, 0.9999\}$. Table 3 summarizes our final choices of these hyperparameters.