# OpenReview forum: "Treepiece: Faster Semantic Parsing via Tree Tokenization"
_EMNLP/2023/Conference — EMNLP 2023 Findings_

### Official Review · Reviewer_JKxP · 2023-07-24

**Soundness:** 4

**Excitement:**

3: Ambivalent: It has merits (e.g., it reports state-of-the-art results, the idea is nice), but there are key weaknesses (e.g., it describes incremental work), and it can significantly benefit from another round of revision. However, I won't object to accepting it if my co-reviewers champion it.

**Paper Topic And Main Contributions:**

This paper is about improving inference speed of semantic parsing models based on autoregressive (AR) encoder-decoder setting.
Due to their autoregressive property, AR models tend to have slower inference speed compared to non-AR models.
At the same time, non-AR approaches perform considerably worse than AR approaches.
To address this issue, the authors propose to tokenize parse trees into larger units, called TreePiece units. As a result, the average length of each logical form becomes shorter and the inference speed could be made faster.
The core part of the work comes from obtaining high-quality TreePiece vocabulary (which is a minimal vocabulary of a Treepiece tokenizer) and TreePiece simplex (probability assignment for each token in TreePiece vocabulary).
Inspired by Byte Pair Encoding, TreePiece vocabulary is obtained by merging high-frequency units, and the simplex is obtained from frequencies of those.
From experiments, the authors empirically prove that the method accelerates the decoding and inference time needed for AR models both in CPU and GPU settings, while maintaining performance degradation minimal.

**Questions For The Authors:**

- Do you have statistics about the average length before and after applying the TreePiece method?
- What do you think of the reasons that NAR methods perform worse than AR methods in semantic parsing tasks?

**Reasons To Accept:**

- The terms and algorithms are thoroughly defined
- Advantages seem to be promising

**Reasons To Reject:**

- Content needed for understanding the paper heavily relies on the one from appendix.

**Reproducibility:**

4: Could mostly reproduce the results, but there may be some variation because of sample variance or minor variations in their interpretation of the protocol or method.

**Reviewer Confidence:**

3: Pretty sure, but there's a chance I missed something. Although I have a good feel for this area in general, I did not carefully check the paper's details, e.g., the math, experimental design, or novelty.

---

> ### Author Rebuttal · Authors · 2023-08-26
>
> Thank you for your constructive feedback. We address each point as follows:
>
> > Content needed for understanding the paper heavily relies on the one from appendix.
>
> We recognize the inconvenience of referring to the appendix for core content. To improve clarity, we'll reorganize the manuscript, integrating essential appendix materials into the main text.
>
> To answer your questions:
>
> > Do you have statistics about the average length before and after applying the TreePiece method?
>
> The following table compared average lengths (measured in generation steps) before and after applying TreePiece (i.e., AR vs. TreePiece-600) for both logical forms and skeletons.
>
> | | |Avg. lengths |
> |---|---|---|
> | TreePiece-600 | Logical form  |$2.2$ |
> | | Skeleton| $1.2$ |
> | AR | Logical form |$9.6$ |
> | | Skeleton |$6.2$ |
>
> Additionally, to showcase TreePiece's efficacy in reducing lengths, we include two more tables below. The first table segments test samples by the number of intents, and the second by number of ontologies, both presenting average lengths within each category.
>
> | #Intents| |1 | 2 | 3 | 4 | 5 |
> |---|---|---|---|---|---|---|
> | TreePiece-600 | Logical form  |$2.1$ | $2.4$| $3.4$| $4.2$| $5.0$ |
> | | Skeleton| $1.1$ | $1.4$| $2.4$| $3.2$| $4.0$ |
> | AR | Logical form |$8.4$ | $15.0$| $21.7$| $29.8$| $37.0$ |
> | | Skeleton |$5.2$ | $10.5$| $15.9$| $21.9$| $27.5$ |
>
> | #Ontologies| |1 | 2 | 3 | 4 | 5 | 6|7|8|9|10|11|12|
> |---|---|---|---|---|---|---|---|---|---|---|---|---|---|
> | TreePiece-600 | Logical form  |$2.0$ | $2.0$| $2.2$| $2.2$| $2.4$ |$2.5$ | $2.7$ | $3.2$ | $3.6$ | $4.1$ | $4.1$ | $4.1$ |
> | | Skeleton| $1.0$ | $1.0$| $1.2$| $1.4$| $1.5$ | $1.7$ | $2.2$ | $2.6$ | $3.1$ | $3.1$ | $3.1$ | $3.1$ |
> | AR | Logical form |$2.0$ | $6.0$| $9.9$| $13.0$|$15.0$| $18.2$ | $21.0$ | $23.0$ | $26.2$ | $29.2$ | $30.9$ | $34.3$ |
> | | Skeleton |$2.0$ | $4.0$| $6.0$| $8.0$| $10.0$ | $12.0$ | $14.0$ | $16.0$ | $18.0$ | $20.0$ | $22.0$ | $24.0$ |
>
> > What do you think of the reasons that NAR methods perform worse than AR methods in semantic parsing tasks?
>
> One of the most significant strengths of AR modeling is its ability to capture temporal dependencies by allowing each token to rely on the entire past contexts. This is essential for NLP tasks because word meanings can be deeply influenced by their preceding context. In contrast, **NAR loses this chain of dependencies**, making predictions without full sequence's context, leading to higher error rates. Another crucial factor is Length Prediction: NAR needs to predict output length, which, if incorrect, can lead to completely wrong outputs.

---

### Official Review · Reviewer_7MJZ · 2023-08-04

**Soundness:** 3

**Excitement:**

2: Mediocre: This paper makes marginal contributions (vs non-contemporaneous work), so I would rather not see it in the conference.

**Paper Topic And Main Contributions:**

The paper presents a novel Tree Tokenization technique called "TreePiece" for semantic parsing. The traditional approach uses autoregressive (AR) encoder-decoder neural networks, which can be slow due to their sequential prediction process. The authors propose TreePiece as a solution to accelerate this process. The main contribution of the paper is to use the TreePiece technique, which tokenizes a parse tree into multiple sub-trees and predicts them simultaneously. This method reduces the number of prediction steps and, thus, accelerates the semantic parsing process.

**Reasons To Accept:**

The novel approach (TreePiece) addresses a significant problem in semantic parsing: the slow speed of autoregressive models. By predicting multiple sub-trees simultaneously, TreePiece greatly reduces the prediction steps and, consequently, the computation time.

**Reasons To Reject:**

The improvement in performance and speed is not substantial. When compared to non-autoregressive (NAR) methods, the system exhibits similar performance but operates at a slower pace

**Reproducibility:**

3: Could reproduce the results with some difficulty. The settings of parameters are underspecified or subjectively determined; the training/evaluation data are not widely available.

**Reviewer Confidence:**

4: Quite sure. I tried to check the important points carefully. It's unlikely, though conceivable, that I missed something that should affect my ratings.

---

> ### Author Rebuttal · Authors · 2023-08-26
>
> Thank you for your feedback.
>
> > The improvement in performance and speed is not substantial. When compared to non-autoregressive (NAR) methods, the system exhibits similar performance but operates at a slower pace
>
> We would like to clarify a few points regarding the perceived performance improvement over NAR:
> 1. The task distribution in TOPv2 is skewed towards simplicity: about 85% of samples are **easy tasks** with only 1 intent, an average of 2.65 ontologies, and a tree-depth of at most 3. As models are trained extensively, performance on these tasks tends to saturate, leading to minimal performance differences between methods.
>
>
> 2. The strength of our TreePiece tokenization lies in handling tasks of higher complexity. We provide two tables below to illustrate this: the first segments test samples by the number of intents, and the second by number of ontologies. Notably, while the performance gap is small for simpler tasks, it widens considerably for complex ones, already reaching improvements of 2% for 2-intent frames and 4% for 6-ontology frames.
>
> | #Intents| |1 | 2 | 3 | 4 | 5 |
> |---|---|---|---|---|---|---|
> | TreePiece-600 | EM |$88.10$ | $83.21$| $69.46$| $46.51$| $0.0$ |
> | | EM-S |$90.01$ | $87.23$| $72.73$| $50.00$| $0.0$ |
> | NAR | EM  |$87.72$ | $80.95$| $67.60$| $23.26$| $0.0$ |
> | | EM-S| $89.65$ | $84.81$| $69.93$| $24.42$| $0.0$ |
>
> | #Ontologies| |1 | 2 | 3 | 4 | 5 | 6|7|8|9|10|11|12|
> |---|---|---|---|---|---|---|---|---|---|---|---|---|---|
> | TP | EM |$85.72$ | $90.37$| $88.27$| $86.36$|$80.33$| $83.22$ | $77.04$ | $68.47$ | $58.64$ | $47.73$ | $48.89$ | $41.47$ |
> | | EM-S |$85.72$ | $91.71$| $90.31$| $89.54$| $84.31$ | $76.86$ | $82.55$ | $74.63$ | $65.45$ | $50.00$ | $51.11$ | $50.00$ |
> | NAR | EM  |$85.51$ | $89.89$| $87.82$| $85.01$| $78.62$ |$79.26$ | $73.25$ | $66.75$ | $56.16$ | $43.18$ | $26.67$ | $0.0$ |
> | | EM-S| $85.51$ | $91.37$| $90.00$| $88.68$| $82.96$ | $82.89$ | $78.65$ | $69.95$ | $63.35$ | $46.59$ | $31.11$ | $0.0$ |
>
>
> 3. Historical improvements on TOPv2 in existing literature are of similar scale, such as [1] with an improvement of ~1.2% and [2] less than 0.25%. The difference between models like RoBERTa base and RoBERTa large is often less than 0.4%. Given this context, our improvements of 0.67% on frames and 0.7% on skeletons are significant.
>
> In light of the above, we believe our method leads to substantial improvements and offers meaningful advantages, especially in handling complex tasks.
>
> We thank the reviewer for the observation. In upcoming versions of our paper, we'll articulate the complexity skew of the TOPv2 dataset more clearly, elaborate on our model's proficiency in addressing complex tasks, and incorporate the stratified results to better support our main claims on performance improvements.
>
> [1] Akshat Shrivastava, Pierce Chuang, Arun Babu, Shrey Desai, Abhinav Arora, Alexander Zotov, Ahmed Aly, Span Pointer Networks for Non-Autoregressive Task-Oriented Semantic Parsing
>
> [2] Akshat Shrivastava, Shrey Desai, Anchit Gupta, Ali Elkahky, Aleksandr Livshits, Alexander Zotov, Ahmed Aly, Retrieve-and-Fill for Scenario-based Task-Oriented Semantic Parsing.

---

### Official Review · Reviewer_5cvo · 2023-08-05

**Soundness:** 3

**Excitement:**

4: Strong: This paper deepens the understanding of some phenomenon or lowers the barriers to an existing research direction.

**Paper Topic And Main Contributions:**

This paper proposes a tree tokenization method that allows autoregressive models to generate one subtree at a time instead of one token at a time. This allows significant speed up during model inference, since the output sequences become much shorter with the proposed tree tokenization.

The authors conducted experiments on TOPv2, a semantic parsing benchmark, and reported much faster decoding time (more than 6x faster than a baseline autoregressive decoder) which is comparable to a non-autoregressive decoder in terms of decoding speed, while maintaining similar levels of accuracy to the autoregressive decoder baseline.

**Questions For The Authors:**

* How much is the proposed tree tokenization method restricted to TOPv2-style semantic representations? (which are based on the notion of intents and slots).
* Can you clarify the difference between AR and TreePiece-0 in Table 1? Is the main difference the utilization of NAR decoding for utterance predictions? There seems to a big speed and accuracy difference between the two.
* Are there any recommendations on setting the number of expansion size? Table 1 seems to suggest that a larger number always leads to better accuracy and faster speed. If the expansion size continues to grow, does the accuracy start dropping? Figure 3 contains TreePiece-2200, but its performance is not reported in Table 1.

**Reasons To Accept:**

* This paper makes a focused contribution on reducing the number of decoding steps required through tree tokenization. The method appears to be novel and sound. The idea of decoding one subtree at a time instead of one token at a time is quite intuitive and likely interesting many researchers in the field of semantic parsing.
* The paper provides empirical evidence that the proposed tree tokenization method support much faster inference comparable to non-autoregressive decoder, while maintaining similar accuracies as the baseline autoregressive decoder.

**Reasons To Reject:**

* The proposed tree tokenization method is only tested on a single semantic parsing benchmark TOPv2 and its corresponding semantic representation. It is not clear whether the method is applicable to other semantic parsing tasks and representations.
* A lot of relevant technical details in this paper is put in the appendix (e.g., the main algorithms), making the main paper somewhat less self-contained.

Edit:
Authors have addressed the first point in their responses (though only theoretically and without empirical evidence).

**Reproducibility:**

3: Could reproduce the results with some difficulty. The settings of parameters are underspecified or subjectively determined; the training/evaluation data are not widely available.

**Reviewer Confidence:**

2: Willing to defend my evaluation, but it is fairly likely that I missed some details, didn't understand some central points, or can't be sure about the novelty of the work.

**Typos Grammar Style And Presentation Improvements:**

Figure 3 is blocking some texts on the left column on page 3.

The NAR utterance decoder does not seem to be fully described in the paper.

---

> ### Author Rebuttal · Authors · 2023-08-26
>
> We appreciate the reviewer’s constructive feedback and thoughtful questions. We address each point as follows.
>
> > How much is the proposed tree tokenization method restricted to TOPv2-style semantic representations? (which are based on the notion of intents and slots).
>
> The proposed TreePiece technique, while tested on TOPv2, is not intrinsically bound to it. Our approach requires **only two** conditions for applicability:
> * The dataset offers a closed vocabulary of ontologies.
> * The logical forms inherently carry a tree structure.
>
> Given these conditions, TreePiece can seamlessly adapt to a broad range of datasets. The following are some examples:
>
> 1. GeoQuery Dataset
> 2. Spider Dataset (text to SQL)
> 3. ATIS (Air Travel Information System) Dataset
> 4. WebQuestions dataset
> 5. SQA (Sequential Question Answering) Dataset
> 6. WikiSQL Dataset
>
> To exemplify with the WikiSQL Dataset:
>
> **Utterance**: List books written by authors from the USA that were published after 2000 and have more than 100 pages.
>
> **Logical form:**
> ```
> [SELECT: books
>     [AUTHOR_ORIGIN: USA]
>     [CONDITION:
>         [PUBLICATION_DATE: [AFTER: 2000]]
>         [PAGE_COUNT: [GREATER_THAN: 100]]
>     ]
> ]
> ```
> TreePiece can tokenize the Skeleton (containing 7 ontologies) of this logical form into three TreePiece units:
> ```
> 1. [SELECT: [AUTHOR_ORIGIN: ][CONDITION: [MISSING][MISSING]]]
> 2. [PUBLICATION_DATE: [AFTER:]]
> 3. [PAGE_COUNT: [GREATER_THAN: ]]
> ```
> This demonstrates the adaptability of TreePiece beyond just task-Oriented semantic parsing. We thank the reviewer for bringing up this important aspect, and we will clarify this in our next revision.
>
> > Can you clarify the difference between AR and TreePiece-0 in Table 1? Is the main difference the utilization of NAR decoding for utterance predictions? There seems to a big speed and accuracy difference between the two.
>
> You've accurately observed that both AR and TreePiece-0 generate one ontology at a time, but there are crucial differences:
> * In AR, logical forms are serialized and tokenized at the word level. Ontologies, utterances, and closing brackets are generated in their serialized order.
> * TreePiece-0, on the other hand, generates all ontologies first (in pre-order) and then completes the utterances using NAR decoding (as you've mentioned).
>
> In light of these, let's try to understand the performance difference:
> * **Skeleton Accuracy**: AR has higher accuracy due to the intermingling of ontologies and utterances during generation, which, compared to TreePiece-0, provides more contextual hints for ontology prediction. However, as TreePiece vocabulary size grows and decoding steps decrease, this advantage diminishes (as shown in Table 1).
>
> * **Speed**: TreePiece-0 is twice faster than AR not just because of NAR-utterance-decoding but also due to its more efficient ontology generation. For example, AR requires two steps to complete an ontology ("[IN: CREATE_CALL" and "]"), whereas TreePiece-0 accomplishes this in one step.
>
> > Are there any recommendations on setting the number of expansion size? Table 1 seems to suggest that a larger number always leads to better accuracy and faster speed. If the expansion size continues to grow, does the accuracy start dropping? Figure 3 contains TreePiece-2200, but its performance is not reported in Table 1.
>
> We have indeed conducted experiments on larger sizes beyond Table 1, including 2200. As the table below shows, accuracy begins to decline once the size exceeds 600.
>
> The primary culprit is OOV (Out-Of-Vocabulary) issues. When we encounter frames in test datasets whose TreePiece units contain unseen MISSING placeholder patterns, accuracy is negatively impacted. This is elaborated on in Limitation Section, Section 2.3.2, and Appendix F.
>
> Theoretically, a larger vocabulary means fewer steps to generate a skeleton, leading to higher overall accuracy. However, increasing vocabulary size means more MISSING patterns must be left out (see Footnote 1), which escalates OOV rates, impacting accuracy negatively. There's a balance point where accuracy is optimized. In our experience, this sweet spot lies around 600.
>
> In general, for any given dataset, it is necessary to conduct comprehensive experiments across a range of vocabulary sizes to identify the optimal setting. Nonetheless, this optimal size has an upper limit and won't increase indefinitely.
>
> We will clarify this for better understanding.
>
> | |TreePiece-600 | TreePiece-700 | TreePiece-800 | TreePiece-1200 | TreePiece-2200 |
> |---|---|---|---|---|---|
> | EM |**$86.86$** | $86.79$| $86.73$| $86.51$| $86.44$ |
> | EM-S |**$89.26$** | $89.20$| $89.13$| $89.05$| $88.99$ |
> | OOV (%) |**$0.12$** | $0.18$| $0.26$| $0.34$| $0.40$ |
>
> > Figure 3 is blocking some texts on the left column on page 3.
>
> Thank you for bringing this layout issue to our attention. We will rectify this in the revised manuscript and ensure that all text is visible.
>
> > The NAR utterance decoder does not seem to be fully described in the paper.
>
> We appreciate your observation. To clarify, our utterance NAR decoding mechanism directly builds upon the standard NAR baseline, as detailed in reference [1]. The decoder consists of multi-layer Transformer architecture and operates as follows:
> After generating Skeleton, the requisite number of utterances are determined, denoted as $N$ (refer to lines 112-116). The NAR decoder's input is formed by concatenating the embeddings of the predicted TreePieces and $N$ mask tokens. Each decoder layer performs self-attention as well as cross attention with encoder hidden states. Lastly the decoder generates utterance predictions at these $N$ masked positions.
>
> We hope this clarifies any ambiguity and will incorporate this expanded explanation in the revised paper for comprehensiveness.
>
> [1] Marjan Ghazvininejad, Omer Levy, Yinhan Liu, and Luke Zettlemoyer. 2019. Constant-time machine translation with conditional masked language models.

---

### Meta-Review · Area_Chair_wKo5 · 2023-09-19

**Recommendation:** 4

**Metareview:**

This article proposes a tree tokenization method that provides a substantial speed up of autoregressive semantic parsing strategies. The method provides a way to do faster autoregressive semantic parsing with comparable accuracy, with comparable speed and higher accuracy compared to non-autoregressive methods.

Reviewers agree that some substantial information is contained in the appendix, but argue that this can be resolved within the extra page for a final revision. One reviewer was a bit more negative than the others, based on a claim that has been verified to be false after inspecting the paper. Since no response from said reviewer was received, this has been taken into account in the recommendation.

---

### Decision · Program_Chairs · 2023-10-07

**Decision:**

Accept-Findings

**Comment:**

This article proposes a tree tokenization method that provides a substantial speed up of autoregressive semantic parsing strategies. The method provides a way to do faster autoregressive semantic parsing with comparable accuracy, with comparable speed and higher accuracy compared to non-autoregressive methods.

Reviewers agree that some substantial information is contained in the appendix, but argue that this can be resolved within the extra page for a final revision. One reviewer was a bit more negative than the others, based on a claim that has been verified to be false after inspecting the paper. Since no response from said reviewer was received, this has been taken into account in the recommendation.